# The World Smallest Plants (*Wolffia* Sp.) as Potential Species for Bioregenerative Life Support Systems in Space

**DOI:** 10.3390/plants10091896

**Published:** 2021-09-13

**Authors:** Leone Ermes Romano, Giovanna Aronne

**Affiliations:** Department of Agricultural Sciences, University of Naples Federico II, 80055 Portici, Italy; aronne@unina.it

**Keywords:** duckweed, Lemnaceae, *Wolffia* sp., space plant biology, astrobiology, bioregenerative life support system (BLSS), biomass

## Abstract

To colonise other planets, self-sufficiency of space missions is mandatory. To date, the most promising technology to support long-duration missions is the bioregenerative life support system (BLSS), in which plants as autotrophs play a crucial role in recycling wastes and producing food and oxygen. We reviewed the scientific literature on duckweed (Lemnaceae) and reported available information on plant biological traits, nutritional features, biomass production, and space applications, especially of the genus *Wolffia*. Results confirmed that the smallest existing higher plants are the best candidate for space BLSS. We discussed needs for further research before criticalities to be addressed to finalise the adoption of *Wolffia* species for space missions.

## 1. Introduction

To date, artificial ecosystems such as the bioregenerative life support system (BLSS) are the expected technology to support long interplanetary missions [1,2]. The BLSSs concept includes several interconnected compartments in which different types of organisms are used to recycle resources and making these available to other compartments of the system [3]. Within space BLSSs, the photoautotrophic compartment only requires light from outside of the system and recycles carbon dioxide, wastewater, and other wastes to produce edible biomass, oxygen, and water used by the astronauts. One such system is MELiSSA (Micro-Ecological Life Support System Alternative), a project of the European Space Agency for life support to space exploration [3]. The MELiSSA loop aims to produce food, oxygen and recycle water and carbon dioxide to sustain astronaut life in Space missions to reduce the initial payload and dependency on the Earth. Although a few common crop species have been successfully grown in space, a smooth transition from an experimental context to cultivation is not a given, and specific methodologies for selecting candidate plants for BLSS were thus proposed [4,5,6]. Space farming places more demands on plants than conventional agriculture due to the extreme condition of the space environment, which require plants to tolerate factors, such as cosmic radiation and the absence of gravity, while at the same time sustaining astronaut life. The best crop plants in space must produce edible biomass in a high-quality, fast and reliable way, without wasting resources on the production of non-edible biomass and while thus maximising resource utilisation [7]. Candidate species for space farming are increasing; plants at a different stage of development are considered; they include leafy greens, microgreens (e.g., *Brassica oleracea, Rumex acetosa, Lepidium bonariense, Coriandrum sativum, Amaranthus hypochondriacu),* fruit crops (e.g., *Fragaria vesca*, *Solanum lycopersicum*), and tuber crops (*Solanum tuberosum*) [8]. These candidate species were chosen from among crop species commonly used on Earth, and current efforts focus on improving these crops with respect to, e.g., small size (dwarf varieties), fast growth and optimal nutrient content for the astronaut diet [9]. A different approach focuses on alternative species not yet widely cultivated on Earth but possessing several attra ctive traits for food production in space. *Arthrospira platensis* (commonly referred to as Spirulina) is a current example. This filamentous cyanobacterium is the most widely cultivated photosynthetic prokaryote and is promoted as a candidate for BLSSs for space missions [3,10]. The present review focuses on additional new candidate crops, with attention to duckweeds (family Lemnaceae), and specifically the possible use of *Wolffia* species as plants to be cultivated in BLSSs in space.

Lemnaceae are flowering plants of the monocotyledon subdivision that populate freshwater ponds and slow-flowing water bodies all over the world. With the support of aerenchyma tissue, they float on water and may also be slightly submerged. Their diminutive size varies from 1.5 cm in diameter (*Spirodela polyrhiza*, Giant Duckweed) to less than 1 mm (*Wolffia angusta*, the smallest angiosperm). Flowering can occur among duckweeds, but most species exhibit vegetative reproduction as their primary propagation method. Vegetative reproduction occurs by budding from meristematic zones inside the frond pockets. Several generations of fronds exist at any given time inside the mother frond; this creates overcrowding and compression of the primordium daughter fronds. As soon as the oldest daughter frond detaches from its mother frond, the remaining daughters experience a phase of arrangement that pushes the oldest primordium towards the frond’s pocket; this will generate a mature daughter frond. (For more details, see [11].)

Based on morphological markers, the family Lemnaceae is composed of two subfamilies, Wolffioideae (members lacking roots) and Lemnoideae (members with a variable number of roots) [12,13].

The family Lemnaceae comprises five genera (*Wolffia*, *Wolffiella*, *Spirodela*, *Lemna*, *Landoltia*) and currently includes 36 accepted species [14,15]. Scientific interest in these plants has been increasing in the last two decades, but the number of publications is not distributed uniformly among these genera. The trend of growth in interest focused on the genus *Lemna* (Figure 1).

Besides the nuanced morphological differences, important traits are common to all Lemnaceae and include small size, structural simplicity, exponential growth and genetical uniformity due to the predominant vegetative reproduction. Such traits have led scientists to suggest species of the family Lemnaceae as suitable candidates for space cultivation [16,17]. Despite the growing number of studies, the resulting knowledge cannot be directly applied to space farming and will need further testing in space-relevant environments. Moreover, and as mentioned above, most of the new scientific interest is focused on plants of the genus *Lemna*. The results do not represent all species in the family Lemnaceae due to a high degree of variability between and within species. Within the family Lemnaceae, the genus *Wolffia* includes eleven species and is well known for featuring both the fastest growing and the most diminutive flowering plants globally [18,19]. In addition to the peculiarity of these records, *Wolffia* species possess numerous traits that make them potentially suitable as a candidate crop for plant-based BLSSs. For instance, the average relative growth rate of *Wolffia* species is higher than that of most angiosperms and possibly even other Lemnaceae [19]. The rootless morphology maximises the harvest index [17]. The nutritional profile of *Wolffia* species has excellent characteristics [20] and is improved by the absence of toxic substances (e.g. oxalic acid) [13]. Moreover, *Wolffia* species are suitable for new biotechnological applications and nutrient removal from sewage water [21,22,23,24,25].

In addition, the positive buoyancy typically exhibited by these plants in water could facilitate the transition between Earth and the space environment where the effect of gravity is absent. We considered that all these traits could pave the way for *Wolffia* species to be suitable for space cultivation.

This work reviewed the primary scientific literature on plants in the family Lemnaceae, with an emphasis on *Wolffia* species. The objective was to synthesise the literature and identify possible bottlenecks for utilising *Wolffia* species in space cultivation in BLSS. We focused our attention on plant biological traits, nutritional features, biomass production, and space applications.

## 2. Plant Morphological and Reproductive Traits

Plants in the genus Wolffia have a cosmopolitan distribution, populating the lentic ecosystems in almost all the continents except the Antarctic and Arctic regions [11]. Wolffia is a genus of plants with 11 species and including both the fastest-growing angiosperm [19] and the smallest flowering plants [19] (Figure 2a). As is the case for other species in the family, plants in the genus Wolffia consist of a single physical unit termed frond, or thallus, and interpreted as a leaf and stem in an embryonic stage of development [13,26]. Fronds of Wolffia species have a globose, ovoid boat shape (Figure 2b). In each plant, the newly formed frond (daughter) develops from meristematic cells in a pocket of the older frond (mother) (Figure 2c).

Colonies of fronds consist of a large number of individuals forming the so-called "clusters". Colonies of Wolffia species rarely consist of more than two visible fronds per individual, while, anatomically, each mature plant is composed of multiple individual fronds at different stages of development (Figure 2c) [27,28]. Species of the genus Wolffia do not develop roots and are recounted to absorb water and nutrients through the underside of their main frond, making the root function redundant [27,28,29]. It is worth mentioning that this redundancy of roots functionalities also occurs in other Lemnaceae. More specifically, in Lemna species, plants under replete nutrient concentrations manage nutrient uptake mostly from leaves surface [30]. Vegetative propagation based on meristematic cells in the pocket of the fronds is by far the most recurrent way of plant reproduction [13]. However, under unfavourable conditions (temperatures below 15 °C or nutrient-depleted substrates), Wolffia species can produce perennating organs termed turions, the formation of which is an alternative strategy to normal frond development [31,32].

In Wolffia species, flowers develop inside a cavity that opens near the median line of the upper frond surface (Figure 2d). Flowers are composed of one stamen with the anther and a pistil within which there is one atropous ovule [13]. Recently, laboratory protocols were developed to control flowering in W. microscopica [33]. Nevertheless, several aspects of the reproductive biology and ecology of Wolffia species are still unexplored. One crucial point is to assess how these plants transfer pollen from one to another. Some authors have hypothesised that the main form of pollination in these plants is by fish, birds, or strong wind [13,34,35], but these dispersal models have not yet been verified. Such a lack of knowledge limits the possibility of increasing the genetic variability of plants for breeding and selection.

Wolffia species reproduce mainly by vegetative propagation. The average relative growth rate is noteworthy. It ranges from 0.155 to 0.559 (day^−1^) but is variable among the species of the genera and different clones within a species [19].

Effective vegetative propagation increases biomass production rate, and cloning maintains genetic uniformity facilitating industrial applications. So far, only one genome has been sequenced (W. australiana) [36,37] further comparison among chloroplast genomes shows that the genus Wolffia possesses the most reduced genome size in Lemnaceae [38]. Genome size varied between 375 Mb in W. australiana to 1881 Mb in W. arrhiza [39]. Different genetic manipulation strategies have been conducted and tested on several Wolffia species (W. australiana, W. globosa, and W. columbiana) [40,41,42]. Such manipulations were not stable, except for those conducted by Khvatkov et al. [43] using W. arrhiza, a producer for recombinant human granulocyte colony-stimulating factor. The stability of transient transformation is crucial for being able to adopt Wolffia for mainstream commercial applications. Recently W. globosa was genetically transformed through transient genetic transformation protocols [44]. The advent of stable genetic modification protocols for the Wolffia species is now accelerating the application of these plants in biopharming, where recombinant plants produce complex molecules and proteins. One such example is hirudin production, a peptide produced by leeches saliva that has anticoagulant properties used by modern medicine to cure different types of thrombosis [45,46].

## 3. Human Nutrition

"Khai-nam" (eggs of the water) is the name for duckweed in many countries of south-east Asia. From samples retrieved in many local markets, these water eggs were mainly *W. globosa*, a species endemic to the region mentioned earlier [47,48]. Numerous studies and research projects have been conducted to deepen knowledge about *Wolffia* and the use of these hydrophytes in human and animal diets [29,49,50].

These tiny water plants attracted the interest of many researchers to find new, alternative sources of protein for human populations whose diets are based mainly on corn and starch. Plants of the genus *Wolffia* have a protein profile (both in terms of quality and quantity) necessary to supplement this type of diet [47,50]. *W. microscopica* showed a total protein content of more than 25% (dry weight) and the highest content of the 17 tested amino acids compared to other species of the family Lemnaceae [50]. According to the World Health Organization, the essential amino acid content is greater than the needs of preschool-aged children. The fat fraction comprises 80% polyunsaturated fatty acids, with a high proportion of n-3 polyunsaturated fatty acids; a balanced ratio of n-3 to n-6 in the human diet has been linked to reducing cardiovascular disease [51]. Plants of *Wolffia* species are rich in macro-and micronutrients, but those are strictly dependent on the cultivation substrate [18]. The nutritional values were extensively studied [48,52]. More recently, Appenroth et al. [20] showed that plants of the *Wolffia* species possess the needed and, in some cases, even more, essential amino acid content than those required by pre-school-aged children, with a protein content varying from 20 to 30% freeze-dried weight. Such results are consistent with those of previous research [53]. Starch, fat, and fibre content were also analysed and varied: 10–20% for starch, 1–5% for fat, and ∼25% for fibre. Although the fat content was low in most species, the quantity of polyunsaturated fatty acid was around 80%, and among these, n-3 polyunsaturated fatty acids were more represented than n-6 polyunsaturated [54,55]. Micro and macro elements content were also analysed and reported to be dependent not only on the growth medium composition and the environmental parameters (light intensity) [56,57] but also on the genetic background of the species studied. Furthermore, carotenoid content has been linked to a stress response in *Lemna* species [56] further studies should also investigate this positive stress response in *Wolffia* species.

Due to the high nutrient content of the genus *Wolffia*, the selection of both species and ecotype requires attention to fine-tune cultivation to specific needs of astronauts in space. Among the eleven *Wolffia* species, *W. microscopica* is reported to possess the most significant potential for human nutrition since it has a fast reproduction time and an excellent nutritional profile [20].

*Wolffia* species grown under optimal conditions also resulted rich in phenolic substances [58], a class of bioactive compounds that plays a crucial role in the defence against biotic and abiotic stress in plants [59,60,61] is also well known to be essential for human wellbeing. *W. globose* plants were also reported to be rich in cobalamin (vitamin B12), in a form that is bioactive and well absorbed by the human body [62].

β-sitosterol and stigmasterol are important phytohormones that play a crucial role in plant resistance to pathogen infection and have been found in the extract of *W. arrhiza.* These compounds have high antimicrobial activity, and, have therefore boosted the interest from pharmaceutical and cosmetic companies as nutraceutical constituents [63].

In addition to the nutritional components, a randomised crossover control trial studied the effect of *W. globosa* on the postprandial and overnight glycaemic response [64] showing that patients with abdominal obesity fed a substitute meal shake made of *W. globosa* had a lower postprandial glucose peak than those provided with iso-carbohydrate/protein/calorie yoghurt shake. Moreover, it returned more quickly to baseline glucose levels regardless of the same carbohydrates content. The authors concluded that *W. globosa* might serve as a plant protein substitute with beneficial postprandial glycaemic effects.

When assessing the possibility of introducing plants of *Wolffia* species as a novel food into the human diet, one of the first steps is to study the possible toxic effects on the human body. The authors evaluated genotoxicity and repeated-dose toxicity of dried *W. globosa*, introducing different percentages of dried plants to the diets of rats. The results showed no harmful effect attributable to supplementation with different doses of *W. globosa* to the rat diet [65]. Further studies aimed at testing the possible cytotoxic effect of duckweed on human cells concluded that none of the duckweed extracts from five analysed genera (including *Wolffia*) possessed any detectable anti-proliferative or cytotoxic effect [66].

A further attractive feature for use in the human diet is that *Wolffia* species, unlike other genera in the family Lemnaceae (i.e., *Spirodela*, *Landoltia*, and *Lemna*), have the benefit of not containing any oxalate in the form of calcium oxalate, considering that this compound might be harmful to humans [67].

## 4. Biomass Production and Waste Recycling

Plants of *Wolffia* species can reach a productivity of 265 tons/ha (10.5 tons/ha of dry weight) of fresh weight calculated annually based on nine months in a non-controlled environment [47]. In controlled environments, clusters can reach a production rate of 86–160 g-wet/m^2^/d for the vegetative fronds and 55–64 g-wet/m^2^/d for the turions, where variations are nutrient-dependent [68].

Generally, for research purposes, duckweed growth is measured as the relative growth rate per day (RGR). Naumann et al. [69] describe the calculation of RGR for different bases, including fresh weight, frond number, chlorophyll content, or carotenoid content. The exponential growth rate of duckweeds is measured through doubling time (DT). Under controlled environmental conditions, the highest RGR (0.559/day) was observed in *W. globosa* and the fastest DT (29.3 h) in *W. microscopica*. It is worth noticing the profound variation in RGR and DT among locally adapted ecotypes of the same species, which complicates comparison among species (e.g. [19]). In a comparative study, the RGR of *Wolffia* species was not statistically different from the other genera of the family Lemnaceae [17]. The exponential growth that characterises these plants is presumably related to their high nutrient uptake of 2–6 g/day of nitrogen assimilation per kilo of fresh mass and 1.64–4.94 g/day of ammonium [70]. The natural capacity of assimilating high quantities of ammonium makes this group of plants ideal for recovering nutrients from waste products. *Wolffia* species can remove 70–80% of the nitrogen and phosphorus in secondary treated sewage effluent in 7 days [19]. More specifically, the possibility of using *Wolffia* species as a viable alternative to recover nutrients from wastewaters and converting the produced biomass to bioethanol has been investigated. Experiments conducted with *W. globosa* show that bioethanol production is achievable [71].

The high efficiency in taking nutrients has also been exploited in phytoremediation practice. Numerous experiments have been conducted in this framework, successfully removing pollutant such as arsenic, nitrate, and phosphate from water [23,24].

In addition to pollutants removal, *Wolffia* species also positively contributed to remediation in total suspended solids, chemical oxygen demand, and biological oxygen demand. Moreover, a 50–90% faecal coliform reduction was measured in waters with duckweed mats [72,73].

It is worth mentioning the symbiotic relationship that this group of plants establishes with microorganisms, such as plant-promoting bacteria, cyanobacteria, and microalgae [74]. These close relationships can boost these plants’ nitrogen assimilation effectiveness to these organisms’ capacity fixating nitrogen [75,76].

## 5. Space Application

The reduced size, clonal exponential growth, high yield, and biomass quality make Lemnaceae one of the most favourable candidate plants for use in BLSS [17,18]. The use of aquatic plants in a controlled ecological life support system (CELSS) is ideal [77], as demonstrated in various experiments [78,79]. In the 1990s, the first successful closed BLSS that operated in microgravity conditions was developed and autonomously grew different aquatic organisms. Among these organisms were flowering plants, including *Wolffia* species, animal organisms (invertebrates), algae, and microorganisms. The long duration of the experiment (4 months) made it possible even for the animals inhabiting the Autonomous Biological System to complete an entire life cycle in space [80]. These findings paved the way for the optimisation of food production in *W. arrhiza*-based bioreactors, combining intensive aquaculture systems and a closed regenerative food loop between fish and aquatic plants [22].

Experimental evidence is available for the feasibility of cultivating *Wolffia* in simulated microgravity, where plants exhibited enhanced growth [81]. In addition to the interaction with altered gravity, a study simulating the effect of cosmic radiation on plant growth showed that heavy ions increased the mortality rate of W. *arrhiza* [82]. Considering that the experiment was not conducted in space but on Earth with direct radiation generated under laboratory conditions, it would be worth verifying the effect of space radiation with experiments on orbital platforms such as the International Space Station (ISS). Studies on plants from the genus *Lemna* showed that radiation at low concentrations is mitigated by the production of antioxidant substances such as flavonoids [83]. The benefit of antioxidant substances in the human diet (especially in a space context) needs to be addressed [56,57]. Further studies would be desirable to verify if similar mitigation processes occur in *Wolffia* species.

In the possible cultivation of *Wolffia* species in BLSS, it is also noteworthy that, unlike other duckweeds, fronds of *Wolffia* species can thrive when either floating or submerged [84]. Such a trait could turn out to be helpful for the cultivation of plants in bioreactors instead of greenhouses for the maximisation of biomass production per volume rather than surface area.

## 6. General Discussion and Critical Insights

Plants of the genus *Wolffia* have been less studied than other species from the family Lemnaceae. Nevertheless, available information indicates that *Wolffia* species are attractive candidate species for BLSS and that future research is warranted to support the adoption of *Wolffia* species for space missions.

In particular, further work is needed on differences among species and locally adapted ecotypes in nutritional value, growth, genetics, and the effect of microgravity. The absence of buoyancy and sedimentation characterises the microgravity conditions, absence of convection, absence of hydrostatic pressure, and free-floating of the liquids in the air (container less float). Particular attention should be focused on understanding the physical and physiological reactions of plants *Wolffia* species in a microgravity context. The results of the new studies should be the base points to design appropriate cultivation hardware.

*Wolffia* species are rootless; consequently, they are different from other species already studied under space conditions. Therefore, to validate the adoption of duckweed based BLSS, specific studies should focus on the physiological aspects of nutrient uptakes under microgravity, hypergravity, and partial gravity levels (Moon and Mars gravities).

The vegetative reproduction that characterises most duckweed is fast and reliable to maintain genetic uniformity and maximise biomass production. However, to fast forward the commercial application on both Earth and space of *Wolffia* species and duckweeds in general, deepening knowledge on inducing flowering and achieving successful sexual reproduction is mandatory. The resulting increase of biodiversity will be essential to start the genetic improvement, employing breeding programs aimed at fine-tuning plants to the specific application requirements.

Plant cultivation in bioreactors could be beneficial for space exploration missions where batches of recombinant *Wolffia* species could be employed to produce complex molecules and nutraceutical constituents.

Additionally, as for other crop species, the symbiotic relationships that plants of the genus *Wolffia* can establish with bacteria and other microorganisms should be better investigated. Further investigation should focus on food safety issues as a possible hazard to crew health. However, the symbiotic relationship could maximise the efficiency in nutrient uptake and complex molecule degradation. These relationships could also be beneficial in building a BLSS, including micro-Angiosperm plants (duckweed) and microalgae as photoautotrophic organisms.

In a scenario of space greenhouses, productivity can be maximised by growing the plants in multiple stacked layers, allowing a vertical cultivation system. The possibility of optimising the whole volume of the growth chambers would be enhanced by exploring the potential to cultivate these plants in bioreactors as those currently employed for microalgae cultivation for space missions.

Finally, in a scenario of space cultivation with fully controlled growth systems, the nutritional content of plants of *Wolffia* species can be modulated by fine-tuning the environmental parameters. The protein and starch content can be adjusted to diet requirements by alternating the production of fronds (richer in protein) with turion (richer in starch).

In conclusion, *Wolffia* species hold many features that make them suitable candidate species for space BLSS. Further studies should focus on the effects of multiple environmental factors (including altered gravity, space radiation, light quality and intensity, nutrient management) on plant reproduction, propagation rate, and nutritional traits. Moreover, feasible applications would benefit from new concepts for efficient cultivation hardware.

## Figures and Tables

**Figure 1 plants-10-01896-f001:**
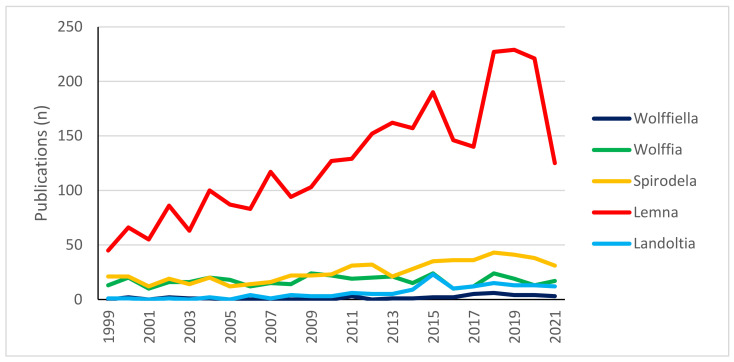
The number of scientific publications per genus of the family Lemnaceae from 1999 to date. Data show the discrepancy between the number of publications for the genus *Lemna* compared to others. Data are from https://www.scopus.com (last searched on 23 July 2021) using the name of each genus as keywords and including only results from 1999 to the date of the search.

**Figure 2 plants-10-01896-f002:**
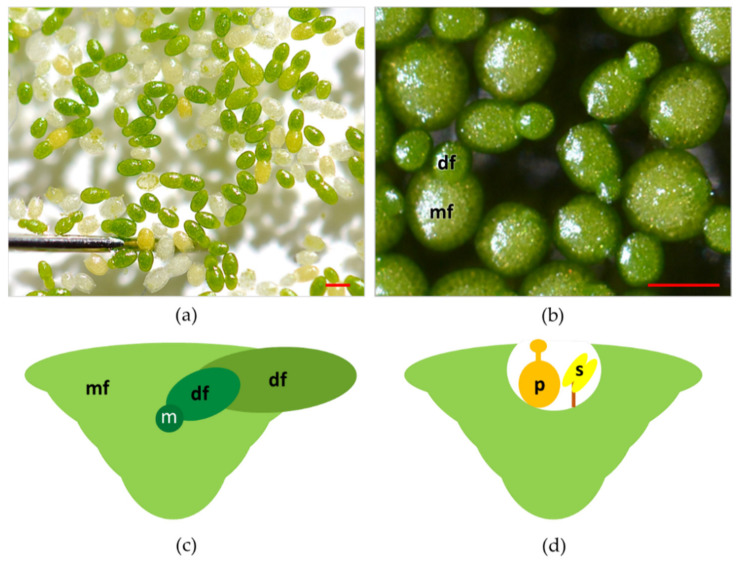
(**a**) Top view of plants of Wolffia arrhiza also showing a stainless-steel pin for size comparison; (**b**) close view of plants of Wolffia arrhiza. Some individual plants feature both the mother frond (mf) and the daughter frond (df); (**c**) schematic transversal section of a plant of Wolffia sp. vegetatively reproducing, mother frond (mf), meristem (m), daughter frond (df); (**d**) schematic transversal section of a plant of Wolffia sp. in full bloom, pistil(p), stamen (s). Red bars = 1 mm.

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
