# Peer review of "The World Smallest Plants (Wolffia Sp.) as Potential Species for Bioregenerative Life Support Systems in Space"

_plants, 2021, doi:10.3390/plants10091896_

Round 1
Reviewer 1 Report
This review addresses the utility of duckweeds of the genus Wolffia as plant candidates for bioregenerative life support systems in space. The review is comprehensive and can serve as a helpful reference for Wolffia biology and applications as well as directions for future research.
General comments:
-The organization of the presentation in the manuscript should be improved (see further details below) and additional key sources integrated in the review
-The manuscript needs to be edited for English usage to improve clarity for the reader. A Word file with extensive edits as tracked changes is provided.
-Further clarification should be given throughout on what is common among duckweeds as opposed to features that are more unique for Wolffia. Any statements about Wolffia having properties that may be even more useful than those of other duckweeds should by more consistent qualitative and especially quantitative comparison between Wolffia and other duckweed species. For example, a comparison of average published growth rates for Wolffia (average across all characterized ecotypes for each species and/or genus) versus other duckweeds and statistical test for significance would be useful. A number of additional examples are highlighted by comments in the edited Word file and include nutritional composition, biomass / protein productivity outdoors (see e.g., Mohedano et al. 2012 Biosource Technology 112, 98 for protein), nitrogen and phosphorus uptake, and production of phenolics, sterols, and other nutraceuticals (see also below).
-For expansion of the section on micronutrients, see also, e.g., Stewart et al. 2020 (Frontiers in Plant Science 11, 480), 2021 (Cells 10, 1481) and/or Demmig-Adams et al. 2020 (Molecules 25, 3607) for additional topics to investigate in Wolffia.
-The rootless nature of Wolffia should be discussed in the context of the fact that Lemna species have rather non-functional roots under high nutrient concentrations where leaves take up the majority of nitrogen (see e.g., Cedergreen and Madsen 2002 New Phytologist 155, 285).
-Expand the section on vegetative propagation a bit.
-The section on plant-microbe interaction should be expanded with citations of available evidence on the effect of microbial partners on duckweed productivity; the endophyte Rhizobium lemnae should also be mentioned and a statement added on whether or not there is evidence for presence of a corresponding endophyte in Wolffia.
-The last section is currently a bit long and repetitive; the authors should condense this section into a Conclusions section and move any new detail into the earlier sections.
Additional comments line by line (see also tracked changes in the attached Word file, including specific suggestions concerning the use of English and editorial comments)
Introduction:
-space should not be capitalized
-bioregenerative should not be capitalized
-line 20-21: why are they most likely?
-lines 21-25: This is not a complete sentence, has grammatical errors., and is too long of a statement. Break into two sentences; first explain what a BLSS is; then, provide MELiSSA as an example.
-Line 27: “recycle” is a better word than restore; “play a crucial role” is unnecessary; furthermore, the compartments transform resources and make them available to other compartments.
-Line 28: This is not necessarily the case. BLSSs have different levels of closure, and only in a closed BLSS can plants do this. For a closed BLSS, more than plants (autotrophic producers) and humans (consumers) may be are needed, such as decomposers to break down organic waste to inorganic nutrients available to plants.
-Lines 30-32: Additional relevant references and examples should be added beyond the two soybean studies cited, to address the extensive space agriculture research conducted to date and reflecting decades of space flight experiments with crop species (especially on the ISS), as well as ground-based experiments through the Russian BIOS experiments, the CELSS program in the US, etc. Review papers on space agriculture research should be cited here as well, such as
Wheeler 2017. Open Agriculture 2(1), 14-32 and Escobar & Nabity 2017 47th International Conference on Environmental Systems.
-Line 30: Plants produce clean water via transpiration in addition to food and O2
-Line 39-41: This sentence does not read well and should be changed to “Candidate species for space farming include...”
-Line 39-41: The authors should provide a list of candidate species as provided by NASA’s Life Support Baseline Values and Assumptions document and the CELSS program, including microgreens, leafy greens, fruiting crops, and tubers. Consider splitting “leafy microgreens” into “leafy greens and microgreens”.
-Line 42-44: An additional trend in genetic engineering of space crops that should be mentioned is to grow dwarf varieties, (for smaller spaces) in addition to increasing nutritional quality and harvest index (production of edible mass).
-Line 76-77: Lemnaceae were considered by NASA in the 1960s, and by others in the 80s-90s, and are currently studied as a crop for space. Examples for sources to cite here for prior research on Lemna or Wolffia as space crops as well as current efforts include Eichhorn & Fritsche 1996 ”Space Station Utilisation 385; Gitelson et al. 2003 Man-made Closed Ecological Systems 9; Gale et al. 1989 Advances in Space Research 9, 43–52; Bluem & Paris 2003 Advances in Space Research 31, 77–86; Escobar & Escobar 2017 47th International Conference on Environmental Systems; Stewart et al. 2020 Frontiers in Plant Science 11, 480; Escobar et al. 2020 International Conference on Environmental Systems.
-Line 52-53: not only ponds but any still or slow flowing freshwater body – streams, ditches, even cracks in cliffs
-Line 77-78: Provide a statement summarizing research questions that need to be answered to utilize the plant for space farming to provide a transition to the last section where recommendations for future research are discussed in detail.
-Line 90-92: Buoyancy does not exist in the absence of gravity and can thus not “facilitate transition” to environments with different gravity. Specifically, because the plants are buoyant, they have adapted to be less gravitropic than land plants, i.e., they are less sensitive to gravity as a cue for growth, which is an advantage in space. The fluid mechanics of duckweed plants in microgravity are an area of current research; see Nabity et al. 2020 International Conference on Environmental Systems regarding surfactant properties and see also Escobar et al. 2020 International Conference on Environmental Systems.
-Line 84-86: A statement made in section 3 (lines 210 – 213) could also be mentioned in introduction, i.e., that, in addition to its small size, remarkable growth rate, and nutritional density, one of the biggest advantages of Wolffia as a space crop over other genera, like Lemna, is the fact that this genus does not produce oxalic acid. Oxalic acid production in genus Lemna was the reason that NASA did not continue their pursuit of duckweed as a space crop in the 1960s. one could also mention here already that rootless growth is another advantage for space crop production systems, making processing (especially harvesting) easier.
Section 2
-Line 154: explain what hirudin is
Section 4. Biomass production and waste recycling
-Line 220: Relative growth rate should not be capitalized
-Line 225: 29.3/h should be 29.3 hours (not per hour)
Section 5. Space applications
-Line 269: Cite here Stewart et al. 2020 Frontiers in Plant Science, 11, 480.
-Line 273: Growth may be limited in submerged Wolffia by dissolved gas concentrations, which is a challenge in microgravity because of diffusion limited gas transport, requiring innovative solutions in growth chamber design and further research.
-Address the possibility of other challenges to establishing space aquaculture for duckweed, such as harvesting (solid/liquid separation challenges), oxygenation of water supply, CO2 supply in the water (gas transport is diffusion limited in microgravity), or if growing on a thin film, maintaining liquid containment.
Section 6.
This is a good list of priorities for research needed to establish feasibility as a space crop. Another challenge for space production is control of algae and other microbial growth (biofilms) in the aquatic system that can compete with duckweed, degrade hardware, and increase risk of pathogens (a hazard to crew health and safety). It will be critical to establish food safety (with respect to pathogens) of duckweed grown in space growth chambers or bioreactors. In addition, the beneficial, microbial partners may have unique responses to the space environment. Wolffia microbiome composition and stability in microgravity is another important area of research.
-Line 333: radiation should be singular (no “s”)

Author Response
Dear reviewer 1
We thank you for the time and the effort spent reviewing our manuscript. We have agreed and accepted virtually all the suggestions and changed the text accordingly. We believe the manuscript is now much improved and hope it is ready to be accepted for publication. Following, we report point by point the replies to your comments. Main changes to the text are highlighted in yellow in the attached Word file. All minor changes have been accepted but not highlighted.
General comments:
-The organization of the presentation in the manuscript should be improved (see further details below) and additional key sources integrated in the review
The manuscript has been changed thoroughly according to the suggestions
-The manuscript needs to be edited for English usage to improve clarity for the reader. A Word file with extensive edits as tracked changes is provided.
The whole manuscript has been edited for English as suggested
-Further clarification should be given throughout on what is common among duckweeds as opposed to features that are more unique for Wolffia. Any statements about Wolffia having properties that may be even more useful than those of other duckweeds should by more consistent qualitative and especially quantitative comparison between Wolffia and other duckweed species. For example, a comparison of average published growth rates for Wolffia (average across all characterized ecotypes for each species and/or genus) versus other duckweeds and statistical test for significance would be useful. A number of additional examples are highlighted by comments in the edited Word file and include nutritional composition, biomass / protein productivity outdoors (see e.g., Mohedano et al. 2012 Biosource Technology 112, 98 for protein), nitrogen and phosphorus uptake, and production of phenolics, sterols, and other nutraceuticals (see also below).
Your multiple suggestions of comparing Wolffia to Lemna are noteworthy. We might consider your suggestions as input to write a new comparative paper, possibly collaborating with other authors expert in Lemna species. Nevertheless, to keep the main aim and structure of this paper unchanged, we included only some of the requested comparisons between Wolffia and other duckweeds species. Please refer to the specific comments below for details.
-For expansion of the section on micronutrients, see also, e.g., Stewart et al. 2020 (Frontiers in Plant Science 11, 480), 2021 (Cells 10, 1481) and/or Demmig-Adams et al. 2020 (Molecules 25, 3607) for additional topics to investigate in Wolffia.
We changed the text and addedthe references to the text.
-The rootless nature of Wolffia should be discussed in the context of the fact that Lemna species have rather non-functional roots under high nutrient concentrations where leaves take up the majority of nitrogen (see e.g., Cedergreen and Madsen 2002 New Phytologist 155, 285).
-Expand the section on vegetative propagation a bit.
The section has been expanded
-The section on plant-microbe interaction should be expanded with citations of available evidence on the effect of microbial partners on duckweed productivity; the endophyte Rhizobium lemnae should also be mentioned and a statement added on whether or not there is evidence for presence of a corresponding endophyte in Wolffia.
The citation has been added. No similar endophyte has been described for the Wolffia genus.
-The last section is currently a bit long and repetitive; the authors should condense this section into a Conclusions section and move any new detail into the earlier sections.
Most of the statements reported in the last section are referred to more than one section above. The other reviewer accepted this layout, and we prefer to maintain this structure unchanged. Nevertheless, we considered your comment and reduced the text to avoid redundancy.
Additional comments line by line (see also tracked changes in the attached Word file, including specific suggestions concerning the use of English and editorial comments)
We modified the text according to suggestions and accepted all minor changes.
Introduction:
-space should not be capitalized
Done
-bioregenerative should not be capitalized
Done
-line 20-21: why are they most likely?
We changed the text
-lines 21-25: This is not a complete sentence, has grammatical errors., and is too long of a statement. Break into two sentences; first explain what a BLSS is; then, provide MELiSSA as an example.
We changed the text according to suggestions
-Line 27: "recycle" is a better word than restore; "play a crucial role" is unnecessary; furthermore, the compartments transform resources and make them available to other compartments.
We changed the text
-Line 28: This is not necessarily the case. BLSSs have different levels of closure, and only in a closed BLSS can plants do this. For a closed BLSS, more than plants (autotrophic producers) and humans (consumers) may be are needed, such as decomposers to break down organic waste to inorganic nutrients available to plants.
We specified this in the text.
-Lines 30-32: Additional relevant references and examples should be added beyond the two soybean studies cited, to address the extensive space agriculture research conducted to date and reflecting decades of space flight experiments with crop species (especially on the ISS), as well as ground-based experiments through the Russian BIOS experiments, the CELSS program in the US, etc. Review papers on space agriculture research should be cited here as well, such as
Wheeler 2017. Open Agriculture 2(1), 14-32 and Escobar & Nabity 2017 47th International Conference on Environmental Systems.
The citation has been added
-Line 30: Plants produce clean water via transpiration in addition to food and O2
This has been included in the text
-Line 39-41: This sentence does not read well and should be changed to "Candidate species for space farming include..."
This has been included in the text
-Line 39-41: The authors should provide a list of candidate species as provided by NASA's Life Support Baseline Values and Assumptions document and the CELSS program, including microgreens, leafy greens, fruiting crops, and tubers. Consider splitting "leafy microgreens" into "leafy greens and microgreens".
We changed the text to improve comprehension
-Line 42-44: An additional trend in genetic engineering of space crops that should be mentioned is to grow dwarf varieties, (for smaller spaces) in addition to increasing nutritional quality and harvest index (production of edible mass).
We changed the text according to the suggestion
-Line 76-77: Lemnaceae were considered by NASA in the 1960s, and by others in the 80s-90s, and are currently studied as a crop for space. Examples for sources to cite here for prior research on Lemna or Wolffia as space crops as well as current efforts include Eichhorn & Fritsche 1996" Space Station Utilisation 385; Gitelson et al. 2003 Man-made Closed Ecological Systems 9; Gale et al. 1989 Advances in Space Research 9, 43–52; Bluem & Paris 2003 Advances in Space Research 31, 77–86; Escobar & Escobar 2017 47th International Conference on Environmental Systems; Stewart et al. 2020 Frontiers in Plant Science 11, 480; Escobar et al. 2020 International Conference on Environmental Systems.
We double-checked and included citations of papers that refer also to Wolffia species.
-Line 52-53: not only ponds but any still or slow flowing freshwater body – streams, ditches, even cracks in cliffs
We changed the text according to the suggestion
-Line 77-78: Provide a statement summarizing research questions that need to be answered to utilize the plant for space farming to provide a transition to the last section where recommendations for future research are discussed in detail.
Unless we misunderstood your comment, we believe in having reported the transitions statements and the aim of the review in the following paragraph at the end of the introduction.
-Line 90-92: Buoyancy does not exist in the absence of gravity and can thus not "facilitate transition" to environments with different gravity. Specifically, because the plants are buoyant, they have adapted to be less gravitropic than land plants, i.e., they are less sensitive to gravity as a cue for growth, which is an advantage in space. The fluid mechanics of duckweed plants in microgravity are an area of current research; see Nabity et al. 2020 International Conference on Environmental Systems regarding surfactant properties and see also Escobar et al. 2020 International Conference on Environmental Systems.
We changed the text to improve comprehension
-Line 84-86: A statement made in section 3 (lines 210 – 213) could also be mentioned in introduction, i.e., that, in addition to its small size, remarkable growth rate, and nutritional density, one of the biggest advantages of Wolffia as a space crop over other genera, like Lemna, is the fact that this genus does not produce oxalic acid. Oxalic acid production in genus Lemna was the reason that NASA did not continue their pursuit of duckweed as a space crop in the 1960s. one could also mention here already that rootless growth is another advantage for space crop production systems, making processing (especially harvesting) easier.
We changed the text according to the suggestion
Section 2
-Line 154: explain what hirudin is
We changed the text according to the suggestion
Section 4. Biomass production and waste recycling
-Line 220: Relative growth rate should not be capitalized
Done
-Line 225: 29.3/h should be 29.3 hours (not per hour)
Done
Section 5. Space applications
-Line 269: Cite here Stewart et al. 2020 Frontiers in Plant Science, 11, 480.
A citation has been added.
-Line 273: Growth may be limited in submerged Wolffia by dissolved gas concentrations, which is a challenge in microgravity because of diffusion limited gas transport, requiring innovative solutions in growth chamber design and further research.
-Address the possibility of other challenges to establishing space aquaculture for duckweed, such as harvesting (solid/liquid separation challenges), oxygenation of water supply, CO2 supply in the water (gas transport is diffusion limited in microgravity), or if growing on a thin film, maintaining liquid containment.
See reply to comment [BD20]
Section 6.
This is a good list of priorities for research needed to establish feasibility as a space crop. Another challenge for space production is control of algae and other microbial growth (biofilms) in the aquatic system that can compete with duckweed, degrade hardware, and increase risk of pathogens (a hazard to crew health and safety). It will be critical to establish food safety (with respect to pathogens) of duckweed grown in space growth chambers or bioreactors. In addition, the beneficial, microbial partners may have unique responses to the space environment. Wolffia microbiome composition and stability in microgravity is another important area of research.
-Line 333: radiation should be singular (no "s")
Done
Commented [BD1]: Clarify. You state below that buoyancy is absent in microgravity.
We clarified the statement according to the comment.
Commented [BD2]: Note that roots are largely not functional in nutrient uptake under replete nutrient concentrations in other Lemnaceae either. Cite Cedergreen and Madsen for their work in showing that nitrogen uptake under high-N conditions occurs mainly through the leaf surface in Lemna, while this shifts under low-N conditions, where roots take up more nutrients than leaves.
We modified the text and added the new citation.
Commented [BD3]: In order to assess whether or not Wolffia species have a signaificantly higher RGRs than duckweeds from other genera, the authors should calculate avearge RGR for all ecotypes within each genus compared, e.g., by Ziegler et al., and do a statistical test on whether or not there is a significant difference between genera. Due to the high degree of variation among ecotypes, this may or may not be the case. Either outcome would be useful to report.
We carefully went through the Ziegler et al. paper. Statistical comparison is reported in the text and consequently we avoided further elaboration of their data. We inserted a new sentence referring to RGRs differences among genera and the new citation in the paragraph 4. (see also comment BD12)
Commented [BD4]: State what that is
We added the statement.
Commented [BD5]: Provide some quantitative information on protein content as well as amino acid composition of Wolffia compared other Lemnaceae. Is there a difference?
We included more information according to the comment.
Commented [BD6]: Provide a source for what ratio of n6 to n3 is beneficial and why.
We clarified the statement according to the comment
Commented [BD7]: Micro is stated twice here. Clarify what is meant by microelements: Mineral elements (?!)
Fat, carbs, and protein are all macro-elements. Add something more on micronutrients. Source 18 lsits caroetnoids. Compare with teh caroetnoid conetnts reported by Stewart et al. 2020, 2021 for Lemna grown under different light intensities.
We changed text and added the new citation.
Commented [BD8]: State how plants were cultivated? Under high light?
High content of phenolic substances was not an effect of light treatment. We modified text for clarification.
Commented [BD9]: State what is known about their function in plants and humans.
We changed text and added the new citation.
Commented [BD10]: Specify what type of yogurt; was the sugar content provided? Was the starch content of Wolffia provide; was it high or low?
We added requested information in the text.
Commented [BD11]: Compare this with reports of Lemna productivity in open ponds; see Modedano et al. 2o12 Bioresource Technology who reported that Lemna produced up to 20x the protein per hectare compared to soybean.
Since both studies have been conducted in open ponds, we consider that a comparison would be misleading due to different environmental parameters.
Commented [BD12]: As suggested above, the authors should calculate is there is a significant difference among species in average RGR of their tested ecotypes.
We added additional information accordingly to both comments.
Commented [BD13]: How does that compare with Lemna? Is it similar or higher for Wolffia?
Please refer to the above general comment.
Commented [BD14]: How does this compare with Lemna?
Please refer to the above general comment.
Commented [BD15]: How does Wolffia compare with Lemna
Please refer to the above general comment.
Commented [BD16]: State which species.
Specific list of species is not reported in the referred citations.
Commented [BD17]: How does Wolffia compare with Lemna? Is ther any indication of an endosymbiont in Wolffia like Rhizobium lemnae in Lemna?
In the referred papers 75, 76 there is no such an indication.
Commented [BD18]: Describe what has been done here; are any similar bioreactor-test available for Lemna for comparison?
Please refer to the above general comment.
Commented [BD19]: Cite here Stewart et al. 2020, 2021 and/or Demmig-Adams et al. 2020
Citations were added
Commented [BD20]: This is an important, attractive point. What is known about CO2, oxygen, and light availability in submerged fronds? Low CO2 availability in water should not be a problem under the elevated atmospheric CO2 levels in space environments?
Indeed this is an open issue to be addressed with new experiments. CO2 levels in space environments (eg. ISS) are significant higher than on Earth.
Commented [BD21]: Eliminate redundant statements; make this a concise conclusions section and move new details up into the earlier sections of the review.
As stated above, most of the statements reported in this section are referred to more than one section above. This layout was accepted by the other reviewer, and we prefer to maintain this structure unchanged. Nevertheless, we considered your comment and adjusted text to reduce redundancy.
Commented [BD22]: Meaning unclear
We chanced the text to clarify the statement.
Commented [BD23]: See comment above about the fact that Lemna roots are not very functional in nutrient uptake under high nutrient levels.
This statement at the conclusion of the paper is aimed to highlight Wolffia traits that can be noteworthy for the use of these plants in space environments.
Commented [BD24]: Condense the reiteration of what has already been siai, move any new detail up to earlier sections.
Please see reply to comment BD21

Reviewer 2 Report
Romano & Aronne survey the potential of the duckweed genus Wolffia as bioregenerative life support system for long-term space missions. They consider it as a suitable candidate, because of fast growth, easy cultivation, optimal biochemical composition and metabolism. Nevertheless, further studies are claimed regarding the optimal species under space conditions and suitable cultivation hardware.
Some minor critical points are:
Line 134: omit 'unfortunately'
Lines 141-3: So far not three Wolffia species have a sequenced genome, but only Wo. australiana (37); the cited references 35, 36 show only sequenced plastoms.
Lines 148/9: transient transformation is per definition not stabile, please remove 'of transient'
Avoid redundancy regarding essential amino acid content (lines 166 and 173); macro- and microelements (lines168 and 179 - here typo!); fat composition (lines 167 and 177/8, explain why different values 60% versus 80%!)
Lines 187/8: it should read: This class of bioactive compound plays a crucial role... and is also...
Line 225: the values 0.559/day for relative growth rate, and 29/h for doubling time need better explanation
Line 273: different (remove 'ly')
Lines 283/4 should read: ... and between different clones of a species...
Check for typos in references 43 (globosa) and 49 (first author)
There is a recent review on all aspects of duckweed research and applications (Acosta et al. Plant Cell 2021) which might be cited in various contexts.
As to the scheme in Fig. 2c: The smallest daughter frond is always the youngest one, so please switch the order of the two df
Author Response
Dear reviewer 2
We thank you for the time, and effort spent reviewing our manuscript. We have agreed and accepted virtually all the suggestions changing the text accordingly. We believe the manuscript is improved and hope it is now ready to be accepted for publication. Following, we report point by point the replies to your comments.
Line 134: omit 'unfortunately'
The word has been removed
Lines 141-3: So far not three Wolffia species have a sequenced genome, but only Wo. australiana (37); the cited references 35, 36 show only sequenced plastoms.
The text has been corrected following suggestions and highlighted in blue in the text.
Lines 148/9: transient transformation is per definition not stabile, please remove 'of transient'
The word has been removed from the text.
Avoid redundancy regarding essential amino acid content (lines 166 and 173); macro- and microelements (lines168 and 179 - here typo!); fat composition (lines 167 and 177/8, explain why different values 60% versus 80%!)
The typo mistakes were corrected.
Lines 187/8: it should read: This class of bioactive compound plays a crucial role... and is also...
It was corrected as was suggested.
Line 225: the values 0.559/day for relative growth rate, and 29/h for doubling time need better explanation
A better explanation was given and highlighted in blue in the text.
Line 273: different (remove 'ly')
"Differently" was corrected to "different."
Lines 283/4 should read: ... and between different clones of a species...
It has been modified as suggested.
Check for typos in references 43 (globosa) and 49 (first author)
Reference has been corrected as suggested.
There is a recent review on all aspects of duckweed research and applications (Acosta et al. Plant Cell 2021) which might be cited in various contexts.
A citation has been added to the manuscript.
As to the scheme in Fig. 2c: The smallest daughter frond is always the youngest one, so please switch the order of the two df
This request is unclear to us. We agree that the smallest daughter frond is always the youngest one, but in Fig. 2c, there is no particular order, and both fronds have the same name (df).

Round 2
Reviewer 1 Report
The authors have addressed most of the suggestions made satisfactorily.
The suggested changes in structure and degree of coverage that were not made are largely a matter of preference.